# Retrospective Case Series Regarding the Advantages of Cortico-Puncture (CP) Therapy in Association with Micro-Implant Assisted Rapid Palatal Expander (MARPE)

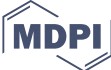

Eugen-Silviu Bud [1] , Mariana Păcurar [1], Alexandru Vlasa [1,*], Ana Petra Lazăr [1], Luminița Lazăr [1], Petru Vaida [2] and Anamaria Bud [1]

1    Faculty of Dental Medicine, George Emil Palade University of Medicine, Pharmacy, Science and Technology of Târgu Mureș, 540142 Târgu Mureș, Romania; Eugen.bud@umfst.ro (E.-S.B.); Mariana.pacurar@umfst.ro (M.P.); Ana.lazar@umfst.ro (A.P.L.); Luminita.lazar@umfst.ro (L.L.); Anamaria.bud@umfst.ro (A.B.)
2    Dudley Group of Hospitals NHS Foundation Trust, Birmingham B18 7QH, UK; p.vaida@nhs.med
*    Correspondence: Alexandru.vlasa@umfst.ro; Tel./Fax: +40-742-825-920

**Abstract:** Transverse maxillary deficiency currently affects 8–23% of adults. One of the most widely used orthodontic treatments today in patients with transverse maxillary defects is the maxillary skeletal expander (MSE). This was a retrospective observational imaging study regarding structural bone changes that may occur during healing after the placement of micro-implant assisted rapid palatal expanders (MARPE) in combination with cortico-puncture (CP) therapy. Regarding the magnitude of the mid-palatal suture opening, the mean split at the anterior nasal spine (ANS) and the posterior nasal spine (PNS) was 3.76 and 3.12 mm, respectively. The amount of split at the PNS was smaller than at the ANS, approximately 85% of the distance, showing that the opening of the midpalatal suture was almost parallel in the sagittal plane. On average, one-half of the anterior nasal spine (ANS) moved more than the contralateral by 0.89 mm. In the present study, we show that MARPE associated with CP therapy had a positive outcome on the midpalatal suture opening. This occurred in safe conditions, without post-surgery bleeding, and showing healing at the corticotomy level, with no signs of swelling or sepsis, which are side effects usually associated with more complex surgical treatments. Our results suggest that non-surgical palatal expansion, assisted by MARPE and CP, is achievable and predictable in young adults.

**Keywords:** palatal split; MARPE; cortico-puncture; orthodontics

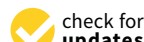

## 1. Introduction

Transverse maxillary deficiency currently affects 8–23% of adults and adolescents [1]. One of the most widely used orthodontic treatments today in patients with transverse maxillary defects is the maxillary skeletal expander (MSE) [2]. In clinical practice, micro-implants were first introduced in the palatal region 25 years ago. The possibility of using these micro-implants at the level of the palatal region is favored by the gingival mucosa at this level having favorable elasticity [3]. With the use of micro-implants associated with maxillary expanders, the incidence of post-interventional complications, such as tooth-borne forces leading to limited skeletal movement and the potential for undesirable tooth movement, root resorption, and lack of firm anchorage to retain sutural long-term expansion, has been reduced [4,5]. The reason is that the micro-implants play the role of the anchoring system for the maxillary expander [6,7]. Transverse jaw deficiency is a common condition in young adults. Untreated or treated incorrectly, it can affect the health of these patients over time. In this sense, modifications of the occlusal plane, damage to periodontal structures, gingival retraction, change in tongue position, asymmetry of the facial planes, and, in severe cases, even sleep apnea syndrome may occur in these patients. Given these

possible co-morbidities, orthodontic treatment is particularly important in these cases and is required in clinical practice [8,9].

To obtain skeletal changes with micro-implant assisted rapid palatal expanders (MARPEs), the applied force should be enough to overcome areas of resistance located in the midface region such as the mid-palatal suture (the first that needs to be disrupted), pterygoid junctions, piriform aperture pillars, and zygomatic buttresses [10]. For this reason, using a MARPE device, force is applied directly into the center of resistance of the maxilla using micro-implants and not the tooth as in classical tooth palatal expanders. As a result of this applied force, buccal tipping of the tooth is prevented, and a more parallel suture opening is promoted [11].

Cortico-puncture (micro-perforation) was introduced in clinical practice as a surgical procedure to shorten orthodontic treatment time. It removes the cortical bone that resists orthodontic force in the jaw and keeps the blood circulation and continuity of bone tissues to reduce the risk of necrosis and facilitate tooth movement [11].

In the present study, we evaluated the treatment outcomes of adult patients with maxillary compression who underwent orthodontic cortico-puncture CP surgery in association with MARPE.

## 2. Materials and Method

In this retrospective study, we assessed the structural changes that occurred during healing after the placement of a micro-implant assisted rapid palatal expander MARPE in combination with cortico-puncture therapy (CP) (Figure 1). Prior to the treatment, cone-beam computed tomography (CBCT) was used to determine bone anatomy and density, palatal suture maturation, and anatomical structures that needed to be avoided during CP therapy. For image acquisition, we used a ProMax 3D CBCT unit (Planmeca, Finland). Images were acquired and saved in JPEG format.

The inclusion criteria were as follows:

- Patients without craniofacial abnormalities;
- No previous orthodontic treatment;
- Transverse maxillary deficiency;
- Unilateral or bilateral crossbite.

Included in the study group were twenty patients aged between 21 and 35 years (mean age 23.8 years). Thirteen were female and seven were male.

With the aid of CBCT, patients were confirmed with stage E sutural fusion as classified by Angelieri et al. [8]. The mid-palatal suture could not be identified, and the parasutural bone density was the same as in other regions of the palate (Figure 2).

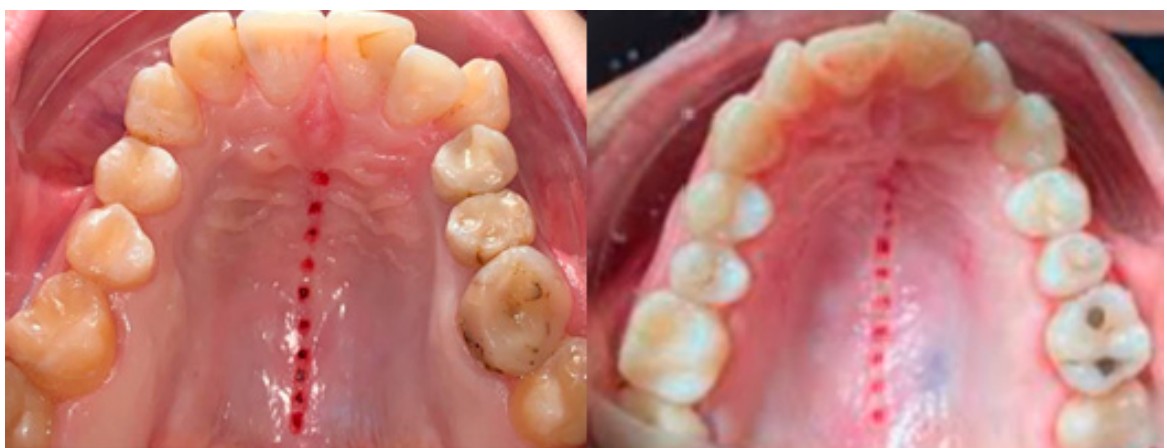

**Figure 1.** Cortico-puncture therapy perforations of the palatal suture.

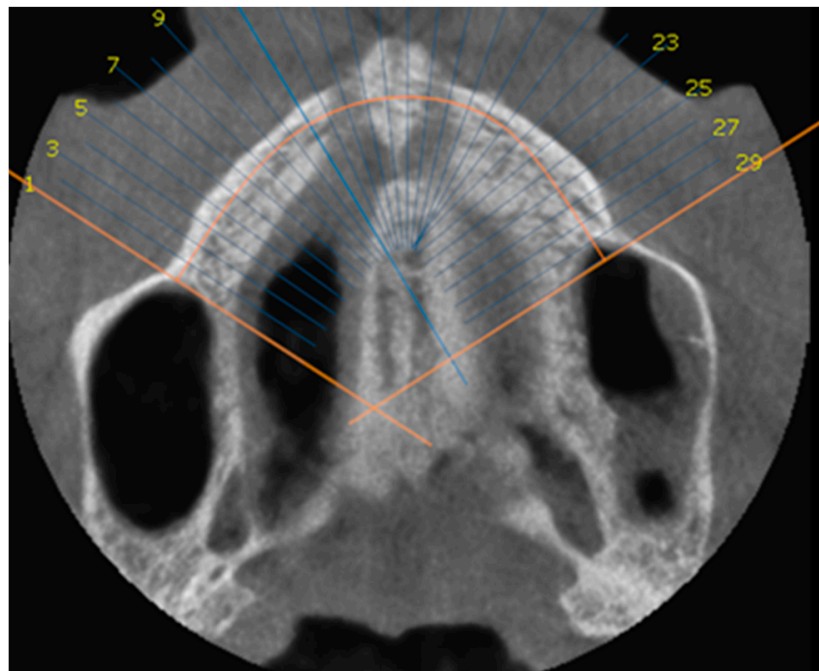

**Figure 2.** Stage E palatal suture as classified by Angelieri et al.

All procedures performed were in accordance with the ethical standards of the Responsible Committee on Human Experimentation (institutional and national) and with the Helsinki Declaration of 1975, as revised in 2008. Informed consent was obtained from all patients for inclusion in the study. The study was approved by the ethics committee of Algocalm Private Medical Center of Târgu Mureș, Romania. (892/5 February 2020).

*Surgical Protocol*

All subjects were treated under the following surgical and orthodontic protocol: Local anesthesia was achieved by applying topical Lidocaine™ Septodont (Creteil, France) 2% spray applied for 1 min, followed by buccal infiltration of the hard palate of a solution of articaine hydrochloride + epinephrine 1:100,000 (ARTICAINE™ Septodont, Creteil, France), administered with the aid of a 0.30 × 38 mm gingival needle (Heraeus™, Hanau, Germany).

Cortico-puncture therapy consisted of the following steps: Ten bone perforations (cortico-punctures) were manually performed along the mid-palatal suture with the aid of a round bur 1.8 mm in diameter and a pilot drill, MIS Implant System™ (Haifa, Israel). Perforations were performed approximately 2 mm apart at a depth ranging from 2–5 mm depending on the thickness of the cortical plate, determined to be safe upon CBCT examination before surgical treatment. After the surgical procedure, analgesic medication (ibuprofen 400 mg 3 × 1/day) was prescribed, and the use of 0.12% chlorhexidine oral rinse for 10–14 days was recommended.

Mini-implant placement consisted of the following steps: After the maxillary skeletal expander II (BioMaterials™, Seoul, Korea) was cemented (Figure 3) in the patient's oral cavity with bands around 1st molars, 4 orthodontic mini-implants (BioMaterials™, Seoul, Korea) (1.8 × 11 mm) were inserted into the palatal bone using the appliance slots as surgical guides. The MSE II was positioned slightly anterior to the soft palate between the first and second molars to direct the expansion forces against the buttress bones.

CBCT images were recorded using a tube voltage of 89 kV and a current intensity of 6 mA, using a cylindrical field of view (FOV) of 82 mm both in diameter and height (Figure 4), the maxilla being our region of interest. The voxel size was 0.2 × 0.2 × 0.2 mm, and the dental arches were positioned similarly in the FOV as presented in Figures 4 and 5.

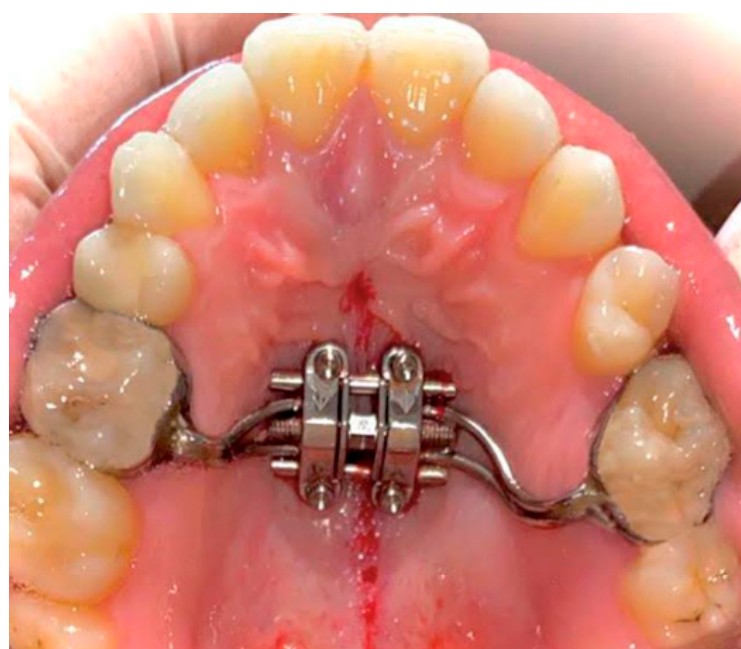

**Figure 3.** MARPE device in position after cementation.

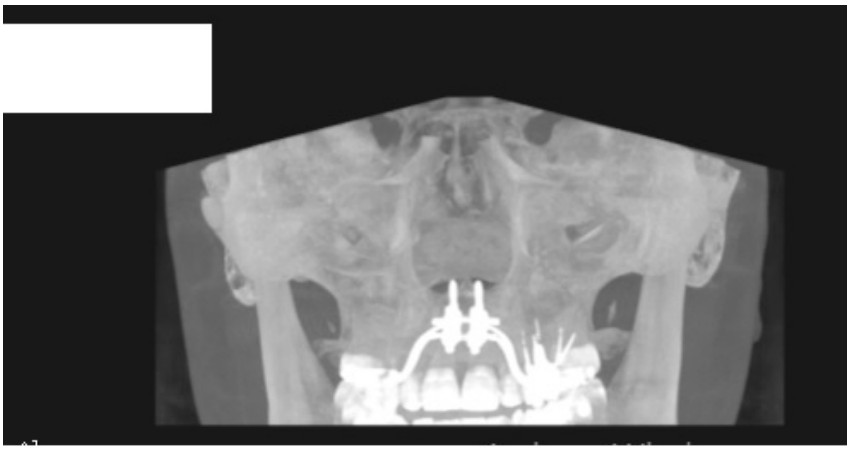

**Figure 4.** The field of view was a cylinder of 82 mm in diameter and height.

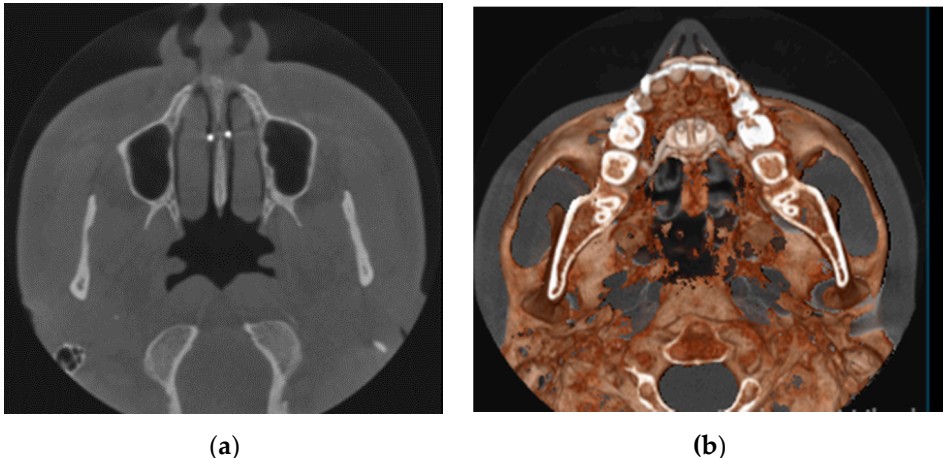

(**a**)　　　　　　　　　　　　　　　　　　　　　　(**b**)

**Figure 5.** Rendering axial images showing FOV images used in the study before the palatal split. (**a**) Micro-implants in position, (**b**) Device in position.

For image acquisition, we used a ProMax 3D CBCT unit (Planmeca, Finland) with the previously mentioned settings. Images were acquired and saved in JPEG format. To locate the cortico-puncture sites, bone modification, and palatal suture split, we used OnDemand 3Ddata App™ software (Seoul, Korea). At each site, our region of interest was a square volume of bone located within the median palatal area (Figure 4). All data recorded were saved using Microsoft Office Excel™, 2017 version, analysis software.

After the surgical procedure, patients were instructed on how to perform the second activation protocol for MSE II: minimum 4–6 turns/day (0.53–0.80 mm/day) until a diastema between central incisors was observed (after 10–19 days), indicating success in splitting the mid-palatal suture (Figure 6), and after diastema appeared: 2 turns/day (0.27 mm/day) until crossbite overcorrection had been achieved. After the expansion, the MSE remained inactivated for at least 2 months to stabilize the expansion.

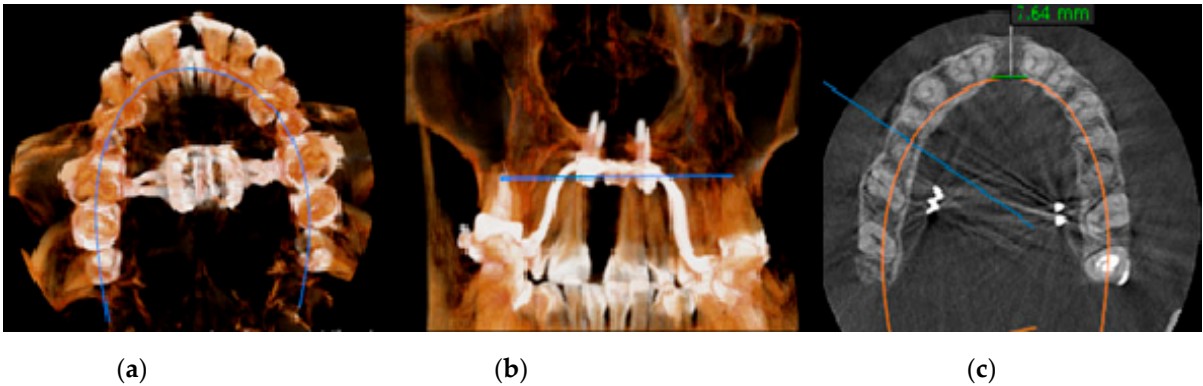

(**a**) (**b**) (**c**)

**Figure 6.** Interincisal diastema formation. (**a**)Axial view (**b**) Frontal view (**c**) Cross-sectional view.

On average, 2 months after the palatal split, new CBCT examinations were performed to assess changes that might have occurred at the skeletal level, such as the dimensions of the anterior nasal spine (ANS) split and posterior nasal spine (PNS) split, interincisal distance split at the cemental-enamel junction (CEJ), and first molar inclination after treatment (Figures 6–11).

To determine the direction of movement that has occurred at the first molar position, a parallel line to the long axis of the tooth was drawn, and the angle between this line and the horizontal plane was measured (Figure 12).

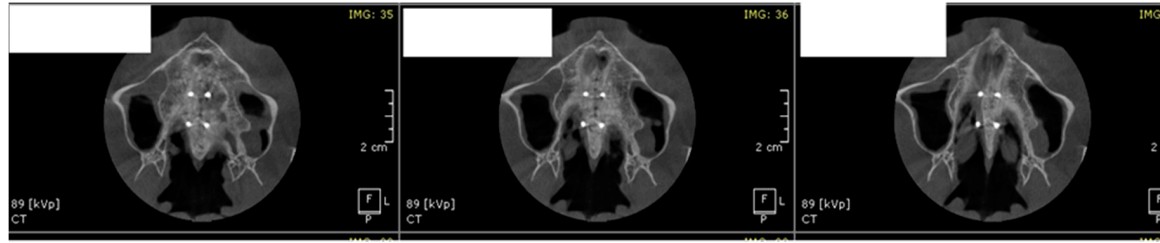

**Figure 7.** Rendering images showing cortico-puncture perforations and micro-implants in position.

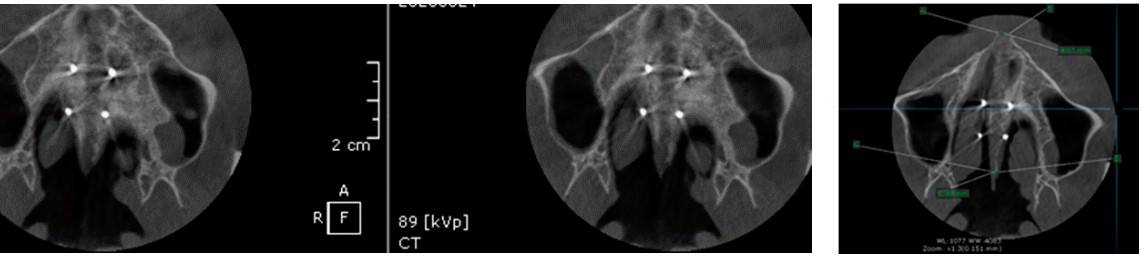

**Figure 8.** The split of the palatal suture.

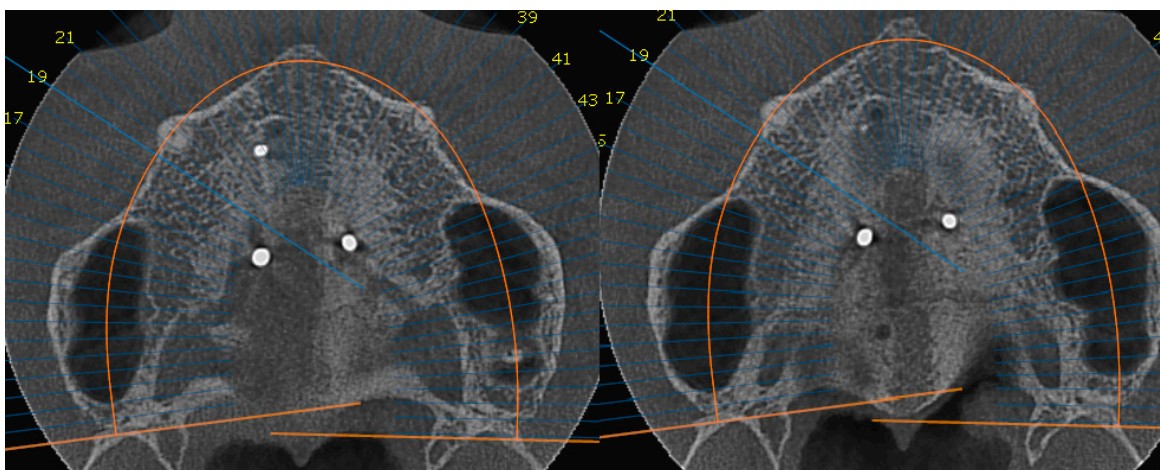

**Figure 9.** The fracture of the palatal plate.

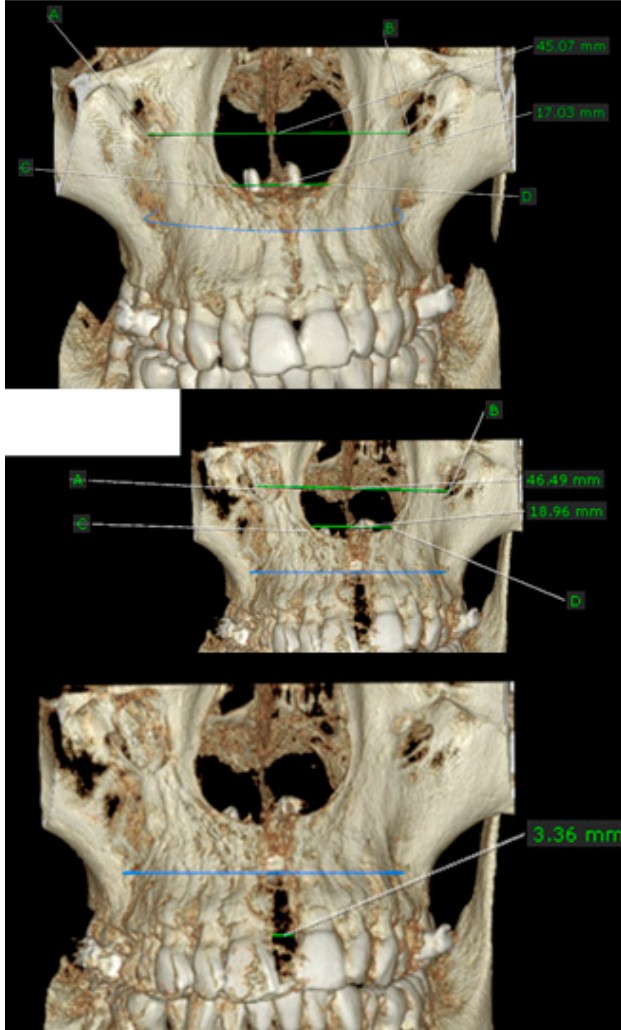

**Figure 10.** Cone Beam Computed Tomography CBCT images used in the study showing different measurements before and after the treatment: A: most mesial point of the infraorbital foramen right side; B: most mesial point of the infraorbital foramen left side; C: most lateral point of the nasal notch right side; D: most lateral point of the nasal notch left side.

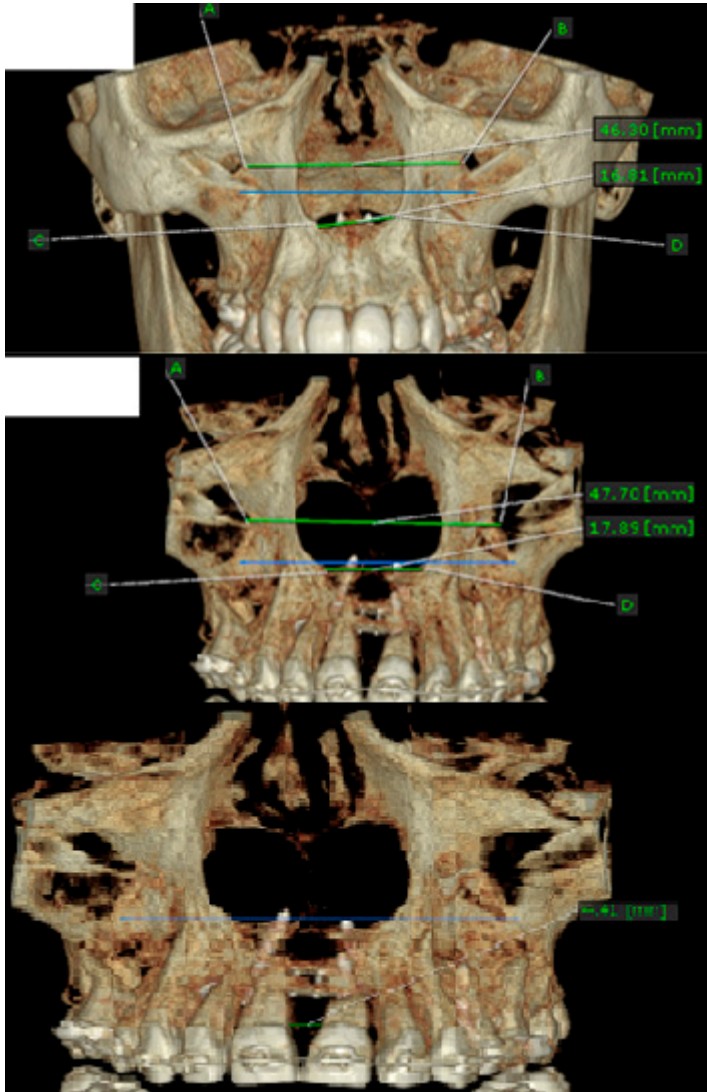

**Figure 11.** CBCT images used in the study showing different measurements before and after the treatment: Measurements on the frontal section on post-expansion CBCT: A: most mesial point of the infraorbital foramen right side; B: most mesial point of the infraorbital foramen left side; C: most lateral point of the nasal notch right side; D: most lateral point of the nasal notch left side.

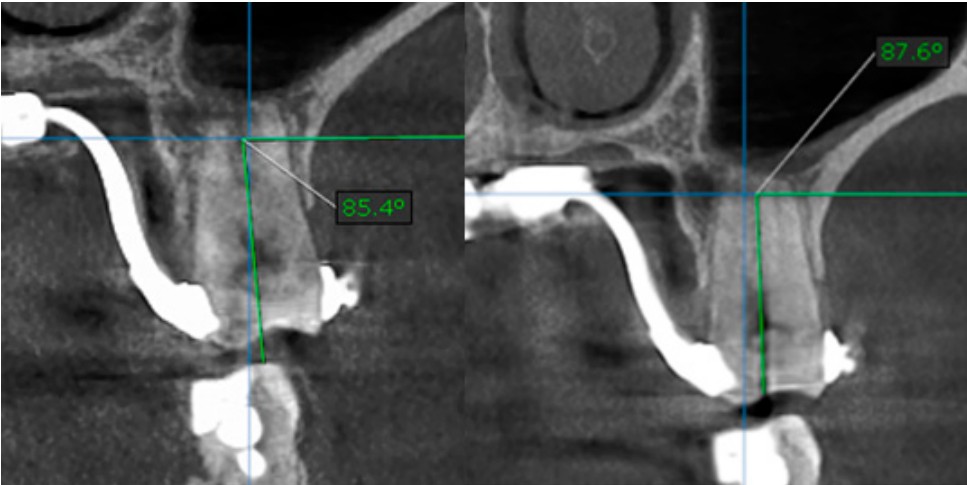

**Figure 12.** The first molar position before vs. after the palatal suture split.

## 3. Statistical Analysis

Statistical processing was performed using GraphPad Prism™ V6.01 software for Windows™. Statistical analysis involved the use of the Student's *t*-test for unpaired (independent) and paired (dependent) data. The D'Agostino and Pearson omnibus normality test was used to determine the normality of the data. The chosen significance threshold was alpha = 0.05, considering *p* significant when $p < 0.05$. A non-parametric test was also chosen because the pre-expansion values of all considered parameters were equal to zero (non-normal distribution).

## 4. Results

Regarding the magnitude of the midpalatal suture opening, the mean split at the anterior nasal spine (ANS) and the posterior nasal spine (PNS) was 3.76 and 3.12 mm, respectively (Table 1). The magnitude of the split at the PNS was smaller than at the ANS by approximately 85% of the distance (Table 2), showing that the opening of the midpalatal suture was almost parallel in the anteroposterior direction. On average, one-half of the anterior nasal spine (ANS) moved more than the contralateral one by 0.89 mm, which was statistically significant ($p < 0.05$).

Regarding the magnitude of interincisal distance at the cemento-enamel junction level, the mean space obtained was 4.10 mm (Table 3) $\pm$ 0.38 standard deviation ($p < 0.05$).

The correlation analysis between first molar tilt, before and after treatment, is given in Table 4. The mean inclination of the first molar was 2.005° SD 0.72° ($p < 0.001$) after treatment, showing that tooth movement also occurred after the splitting of the palatal suture.

**Table 1.** The magnitude of the mid-palatal suture opening in millimeters.

| | Anterior Nasal Spine (ANS) Opening after Treatment | Posterior Nasal Spine (PNS) Opening after Treatment |
|---|---|---|
| Number of values | 20 | 20 |
| Minimum | 2.970 | 2.670 |
| 25% Percentile | 3.250 | 2.780 |
| Median | 3.900 | 2.995 |
| 75% Percentile | 4.165 | 3.385 |
| Maximum | 4.540 | 4.110 |
| Mean | 3.766 | 3.125 |
| SD | 0.4956 | 0.4070 |
| Standard error of mean (SEM) | 0.1108 | 0.09102 |
| Lower 95% CI of mean | 3.534 | 2.934 |
| Upper 95% CI of mean | 3.998 | 3.316 |

**Table 2.** Comparison of the magnitude of the split at the PNS and ANS.

| Table Analyzed | Data 2 |
|---|---|
| Column B | Posterior nasal spine (PNS) opening after treatment |
| vs. | vs. |
| Column A | Anterior nasal spine (ANS) opening after treatment |
| Paired *t*-test | |
| *p*-value | 0.0008 |
| *p*-value summary | |
| Significantly different? ($p < 0.05$) | Yes |

**Table 2.** *Cont.*

| Table Analyzed | Data 2 |
|---|---|
| One- or two-tailed *p*-value? | Two-tailed |
| *t*, df | *t* = 3.964 df = 19 |
| Number of pairs | 20 |
| How big is the difference? | |
| Mean of differences | −0.6410 |
| SD of differences | 0.7232 |
| SEM of differences | 0.1617 |
| 95% confidence interval | −0.9795 to −0.3025 |
| $R^2$ | 0.4526 |
| How effective was the pairing? | |
| Correlation coefficient (r) | −0.2770 |
| *p*-value (one tailed) | 0.1185 |
| *p*-value summary | Ns |
| Significant correlation? (*p* > 0.05) | Yes |

**Table 3.** The magnitude of the interincisal distance opening at the cemento-enamel junction (CEJ) level.

| Parameter | Interincisal Opening at the CEJ Level/Millimeter |
|---|---|
| Minimum | 3.360 |
| 25% percentile | 3.783 |
| Median | 4.210 |
| 75% percentile | 4.365 |
| Maximum | 4.770 |
| Mean | 4.103 |
| SD | 0.3881 |
| SEM | 0.08679 |
| Lower 95% CI of mean | 3.921 |
| Upper 95% CI of mean | 4.284 |

**Table 4.** Changes at the first molar position after treatment.

| D'Agnostino & Pearson Omnibus Normality Test | 1st Molar Tilt after Treatment | 1st Molar Tilt before Treatment |
|---|---|---|
| Number of vales | 40 | 40 |
| Minimum | 85.60 | 84.00 |
| 25% Percentile | 87.60 | 85.23 |
| Median | 88.00 | 85.65 |
| 75% Percentile | 88.65 | 86.70 |
| Maximum | 89.30 | 87.90 |
| Mean | 87.88 | 85.87 |
| Std. Deviation | 1.001 | 1.057 |

**Table 4.** *Cont.*

| D'Agnostino & Pearson Omnibus Normality Test | 1st Molar Tilt after Treatment | 1st Molar Tilt before Treatment |
| --- | --- | --- |
| Std. Error of Mean | 0.2238 | 0.2363 |
| Lower 95 % CI of mean | 87.41 | 85.38 |
| Upper 95 % CI of mean | 88.34 | 86.36 |

| Table Analyzed | Data 1 |
| --- | --- |
| Column B | 1st Molar tilt before treatment |
| Vs. | Vs. |
| Column A | 1st Molat tilt after treatment |
| Paired *t* test | |
| *p* value | < 0.0001 |
| *p* value summary | |
| Significantly different? (*p* < 0.05) | yes |
| One- or two-tailed *P* value | Two-tailed |
| *t*, df | *t* = 12.33 df = 19 |
| Number of pairs | 40 |
| How big is the difference | −2.005 |
| SD of differences | 0.7273 |
| SEM of differences | 0.1626 |
| 95% confidence interval | −2.345 to −1.665 |
| R square | 0.8889 |

## 5. Discussion

As described in the literature, tooth-borne expanders produce a V-shaped opening of the midpalatal suture, with the greatest opening anteriorly and progressively less separation toward its posterior part [12–15]. Lione et al. [14] used a tooth-borne maxillary expander activated by 7 mm on all patients and found that the opening of the midpalatal suture was 3.01 and 1.15 mm at the ANS and PNS, respectively. Conversely, in the patients treated with MARPE in our present study, the borders of the midpalatal suture moved almost perfectly parallel to each other, as evidenced by the amount of split at the PNS, which was 85% of that at the ANS.

Although some authors recommended classic orthodontic treatment in young adult patients with transverse maxillary deficiencies, the failure rate in such cases can be high [16]. Performing an initial cone-beam computed tomography (CBCT) examination is crucial in limiting therapeutic failure in patients undergoing MARPE. The advantage of this method is that it provides important details about the local anatomy, the thickness of the palatine bone, and anatomical details about the articulation of the palatine bone with the pterygoid process to the sphenoid bone. This allows proper planning of this orthodontic treatment. Knowledge of local anatomical details plays a particularly important role in these cases [17,18].

Analyzing the transverse asymmetry of the mid-palatal split (Table 1), on average, one-half of the ANS moved more than the contralateral one by 0.89 mm. The reason for this is unclear but could be related to external forces, such as the presence of a unilateral crossbite that hampers the movement of one maxilla.

Lin et al. [19] published a comparative study between tooth-borne and bone-borne MARPE on late adolescents. The MARPE used in their study included four micro-implants embedded in two acrylic shelves supporting the jackscrew. All implants were positioned close to the dentition, inferiorly from the mid-palatal suture, but the appliance did not

contact the dentition. Angular measurements were performed to assess alveolar bone bending and dental tipping using an arbitrary palatal plane. Significant alveolar bone bending and dental tipping even with this bone-borne expander treatment were found. The dentoalveolar changes may have been possible because of the force applied to the dentoalveolar region by this appliance; however, it is difficult to accept that dental movement could occur even though the expander did not have any physical contact with the dentition. This implies that angular measurements from arbitrary points cannot accurately assess the true impact of an appliance.

The activation force of a rapid maxillary expander (RME) device initially results in the compression of the periodontal ligament, bending of the alveolar bone, and tipping of the anchored teeth [20]. Therefore, a 1–24° increase in molar inclination is inevitable, probably because of alveolar bending and/or tipping of the posterior teeth [21]. Winsauer et al. [22] showed this in a study of 33 adults aged between 23 and 33 years, where 90% of the patients had successful palatal widening without a surgically assisted rapid palatal expander (SARPE) and no dental side effects [22].

Moon et al. [23] conducted a study on 48 late-adolescent patients divided into two groups according to the type of expander: MSE group ($n$ = 24, age = 19.2 $\pm$ 5.9 years) and bone-borne expanders group ($n$ = 24, age = 18.1 $\pm$ 4.5 years). CBCT scans were taken before treatment and 3 months after expansion. Transverse skeletal and dental expansion, alveolar inclination, tooth axis, buccal alveolar bone height, thickness, dehiscence, and fenestration were evaluated on the maxillary first molar. They concluded that the incorporation of teeth into bone-borne expanders resulted in an increase in the severity of side effects. For patients in late adolescence, tissue bone-borne expanders offer comparable skeletal effects to tooth bone-borne expanders, with fewer dentoalveolar side effects [23].

In a randomized control trial regarding the use of bone-borne expansion in the adolescent population, Celenk-Koca et al. [24] showed an increased extent of skeletal changes in the range of 1.5–2.8 times that of tooth-borne expansion and did not result in any dental side effects [24]. In another study of late-adolescent patients, Liu et al. showed that bone-borne expanders produced greater transverse skeletal expansion than tooth-borne Hyrax expanders. In the bone-borne expander group, there was less alveolar bending, less dental tipping, and less vertical alveolar bone loss at the first premolar [19].

Regarding the limitations of MARPE devices, Garib et al. reported that MARPE therapy requires a longer activation time and twice the force for the rupture of the medial palatine suture compared to SARPE [25]. Other disadvantages of MARPE: it may cause temporary inflammation of the palatal mucosa [26], difficulty in hygiene around micro-implants, and risk of infection [1]. Regarding other limitations, Choi et al. named the possibility of failure to separate the suture due to the resistance of the craniofacial structures [27] and when a patient presents a narrow and deep palate because the proper position of some MARPE devices cannot be achieved, since it should be placed close to the palatal mucosa [28]. In such cases, surgically assisted rapid palatal expanders (SARPE) is often suggested to these patients. This surgical procedure increases expansion predictability and success and reduces its side effects. One of the available SARPE techniques consists of a LeFort I osteotomy, associated with surgical rupture of the mid-palatal suture, which decreases mechanical resistance to the lateral forces that will be applied by Hyrax expanders, usually anchored to the first molars and first premolars. However, despite its benefits, SARPE requires hospitalization and increases the biological and financial costs of the treatment [16].

To minimize the surgical procedure and reduce postoperative discomfort, other techniques may be recommended. Cortico-puncture is a method that increases the expression of cytokines and chemokines responsible for stimulating the differentiation of osteoclasts in bone remodeling and thus increasing the rate of tooth movement by up to 62% [23].

Our study demonstrates the use of CP therapy as a surgical method for accelerating bone remodeling to complement the MARPE technique to facilitate suture split. In their study on maxillary expansion in rabbits, Pulver et al. suggested that greater skeletal ex-

pansions may be possible when combined with surgical methods such as cortico-puncture to promote regional acceleratory phenomenon (RAP), stimulating bone remodeling, and reducing bone volume and density [29]. In animal studies, Tsai et al. [30] compared the effects of corticotomy and bone microperforations and concluded that both techniques increased bone remodeling, and there were no significant differences between them. Suzuki et al. [31] demonstrated that when performed along the midpalatal suture, a minimally invasive surgical method such as CP accelerated bone remodeling and favored suture split when this failed to occur after the conventional protocol for MARPE activation [31].

Regarding stability after splitting in literature, Choi et al. [27], using MARPE, found an 86.96% success rate in young adult patients (mean age = 20.9 ± 2.9 years), with stable results after 30 months of follow-up.

For the treatment of transverse maxillary deficiency in adults, Hassan et al. reported that assisted expansion with corticotomy, defined as decortication on the buccal and palatal walls of the alveolar bone, is an effective technique and suggested that the technique may provide greater stability and better periodontal health than conventional expansion. However, the same study reported that there might be side effects of the corticotomy method, such as mild bone loss and loss of inserted gingiva [32]. Studies recommend the use of bone grafts to conserve the periodontium to avoid this [33]. In addition, subcutaneous hematomas and postoperative swelling and discomfort were also associated with the corticotomy procedure [32].

Our study sample comprised only 20 young adults and has a high risk of bias, which emphasized the necessity of further prospective studies involving larger numbers of patients and longer-term evaluation of stability and periodontal adaptation after MARPE.

## 6. Conclusions

1. MARPE associated with cortico-puncture therapy efficiently split the midpalatal suture in adults. The mean split at the anterior nasal spine (ANS) and posterior nasal spine (PNS) was 3.76 and 3.12 mm, respectively. The magnitude of the split at the PNS was smaller than at the ANS (by approximately 85% of the distance), showing that the opening of the midpalatal suture was almost parallel in the anteroposterior direction.
2. MARPE therapy associated with cortico-puncture therapy had a positive outcome on midpalatal suture opening and maxillary advancement, but a medium molar inclination of 2.005° was also observed, suggesting that tooth movement cannot be avoided because of the anchorage of the MARPE device at the molar level.
3. Our results suggest that non-surgical palatal expansion, assisted by micro-implants and cortico-puncture, is achievable and predictable in young adults. This occurs in safe conditions without the need for more complex surgical treatment.
4. The combination of MARPE and the cortico-puncture method proved to be a non-surgical treatment option to correct maxillary transverse deficiency in young adult patients. Cortico-puncture was able to weaken the suture interdigitation, thus facilitating the split.

**Author Contributions:** All authors contributed equally to this research. E.-S.B., M.P. and A.B. designed and performed the surgical phase. A.P.L., P.V. and L.L. derived the models and analyzed the data. A.V., E.-S.B. and M.P. assisted with measurements and helped carry out the statistical analysis. A.V., A.P.L. and L.L. produced the manuscript in consultation with E.-S.B., P.V. and A.B. All authors have read and agreed to the published version of the manuscript.

**Funding:** This research received no external funding.

**Institutional Review Board Statement:** The study was conducted according to the guidelines of the Declaration of Helsinki, and approved by the Ethics Committee of Algocalm Private Medical Center, Targu-Mures, Romania (892/5 February 2020).

**Informed Consent Statement:** Informed consent was obtained from all subjects involved in the study.

**Conflicts of Interest:** The authors declare no conflict of interest. Authors declare that they have no conflict of interest regarding this manuscript and we did not receive any financial support from an organization or a research grant.

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
