# Peer review of "Retrospective Case Series Regarding the Advantages of Cortico-Puncture (CP) Therapy in Association with Micro-Implant Assisted Rapid Palatal Expander (MARPE)"

_applsci, doi:10.3390/app11031306_

Round 1
Reviewer 1 Report
Summary:
This is the resubmission of originally rejected article after some changes. The authors aimed to discuss their findings of a non-surgical approach to palatal expansion using microimplants and cortico-puncture in adult patients.
General Comments:
While this paper has some interesting results and CBCT imaging demonstrating parallel opening of the palatal suture in response to CP and MARPE treatment, it should be re-classified as a case study. There weremultiple issues in the original submission that should be addressed before consideration for publication. In the resubmission, the authors add some information to justify their studies, however, the major issues have not addressed at all.
There is no control mentioned, and there are numerous errors and clarifications that should be addressed before consideration for publication. The introduction and discussion should be rewritten as to provide rationale for the clinical significance of the study and the impact it has on practice.
Additionally, the title of the paper does not support the content of the paper. This should perhaps be revisited again. The revised title has a fatal spelling error.
The authors should re-organize their images and tables so that the results are clearer. The authors also need to address their inclusion/exclusion criteria in the methods.
Specific Comments:
1. First and foremost, there are still numerous spelling errors, grammatical errors, tense errors and article use errors throughout the paper even in the resubmitted version.This reviewer is reluctant to list all necessary issues to be corrected, but below are some examples:
Title: Advanges should be Advantages
Line 77: Round is spelled incorrectly
Line 78: Approximately is spelled incorrectly
Line 82: Recommended is spelled incorrectly
Line 116: Skeletal is spelled incorrectly
The paper should be carefully read and inspected to address these and the remaining issues within the paper.
2. There are phrases used that lack explanation.
Line 23: This occurred in safety conditions, without the need of more complex surgical treatment.
The authors should clarify what is meant by safety conditions, as this is not an objective description.
3. Page 1, Line 27: Transverse maxillary deficiency can be treated differently for prepubertal, circumpubertal and post-pubertal patients.
This is not clear based on the introduction. Please include a sentence with references about the different treatments for these different groups of patients based on where they are in their growth cycle.
4. Page 2, Line 55-58: The inclusion and exclusion criteria need more description and should be rephrased.
Points to address:
- How much transverse discrepancy existed among these patients?
- Was there any history of orthodontics?
- How are healthy subjects defined
- How is poor oral hygiene defined
5. In Figure 3 it appears that copious amounts of cement were left on the teeth after placement of the expander.Is this representative of all images? Was this much cement indeed left on the teeth?
If so, this could impact the inclination results, as it appears that the patient’s would potentially have premature occlusion on those teeth.
6. Some of the images have brackets and some do not.The authors should address how this would impact results.
Mainly, why do some patients have fixed appliances and why do others not? How would the continuous wires passing through to the molars impact the molar inclination?
7. When reading the results, the authors compare ANS opening and PNS opening before and after treatment, which does not indicate that there is any control.
8. Page 11: There should be references to support the statements in line 183-176.
9. Page 12: The discussion does not include any reference to the use of cortico-puncture. As this is a central conclusion that the authors make (Conclusion 3 Line 239 and Conclusion 4 Line 243), there should be ample discussion around the rationale and the clinical significance of this.
Reviewer 2 Report
This study is a very interesting study. However, followings should be considered.
- The introduction did not deal with the background and the need for the present study. There should be a logical sequence leading to the purpose of the study.
- Why and when were informed consents obtained although this study was a retrospective study.
- Mean age of the study group should be described.
- Anterior and posterior limits of cortico-punctures should be described. Was incisive canal protected from the cortico-punctures?
- What is the difference among the images of Figs 8 and 9? Can they be merged into one figure with three images?
- Statistics should be more simplified. Too many unnecessary data. Focus on the statistical differences.
- The rationale for the cortico-puncture should be explained. No references for cortico-puncture.
- It seems that the authors believe that the resistance at the midpalatal suture is the main resistance to the RME. Stanley Liou suggested that midpalatal suture always opens when the lateral maxillary osteotomies are performed. Therefore, the hypothesis of authors has very low possibility. It seems that the main resistance is in the other sutures like pterygopalatine suture, zygomaticotemporal suture, etc.
- To prove the effect of cortico-puncture, the comparsion between two MARPE groups with cortico-puncture and without corticopuncture should be compared. Otherwise, this study only proves that cortico-puncture do not interfere with the MARPE, and this is meaningless.
- Typos were checked on the attached pdf.

Reviewer 3 Report
The authors performed a retrospective study that used orthodontic treatment (maxillary skeletal expander; MARPE in combination with cortico-puncture therapy) for the treatment of patients with transverse maxillary defects. The study focused on analyzing the structural changes of bone that may occur during healing. The mean split at the ANS and PNS were measured. The authors concluded that the combinatorial treatment of MARPE with CP showed positive outcome mid-palatal suture opening and maxillary advancement without the need for additional complex surgical treatment and this treatment modality is predictable in young patients. Although, the manuscript discusses a significant clinical problem, needs rigorous revision based on the following comments.
Introduction:
- Make sure the abbreviation for MARPE is added here first instead of at the start of materials and methods.
Materials & Methods:
- The title should read “materials” instead of “material”
- Line 48: Unnecessary “(“ in between retrospective / and
- Line 48: It would be better to reword it as “This retrospective studied look at structural changes…”
- The second picture is figure 1 is blurry- Please replace it with a clearer image
Surgical protocol:
- Local anesthesia application is a little unclear. Did the patients receive buccal infiltration of anesthesia for palatal surgery?
- Line 74: “In the mucosal area only” indicates either buccal infiltration or very posterior palatal delivery (i.e. into the soft palate).
- Please describe the anesthesia delivery using the terminology in Malamed’s textbook for anesthesia
- Line 76: Re-word “Ten bone perforations, called cortico-punctures, were performed…” to “Ten bone perforations (cortico-punctures) were performed to…”
- Line 77: Is this a typo? “roung” should say “round” I believe
- Please indicate the size of the round bur
- Please indicate how the depth of the puncture was determined.
- Was the depth based on the thickness of the cortical plate?
- Please reword: “Perforations were made at approximately 2 mm apart at a depth ranging from 2 mm to 5 mm, depth appreciated to be safe on CBCT examination before surgical treatment”
- Already stated that CBCT was used to evaluate anatomical structures previously.
- What analgesic medication was prescribed? For how long?
- Please reword: “Oral rinse based on chlorhexidine” to “0.12% chlorhexidine oral rinse”
The mini-implant procedure consisted of the following steps:
- Recommend rewording title to “Mini-implant placement” to match previous “Surgical protocol” title
- Where was the appliance cemented? (i.e. was it held in with bands around molars?)
- Recommend changing “(1.8 mm diameter and 11 mm long)” to “(8 mm by 11 mm)”
- Line 88: Please describe the direction of force (i.e. at what angle with respect to palatal vault or long axis of teeth)
- Lines 91-93: A little confusing. Please provide a figure outlining these measurements.
- Line 94: Please clarify if this is a post-surgical CBCT. If so, how long after surgery was this taken?
Figure 5: Is it possible to outline where the square voxel was taken from?
- Line 112: and after diastema appeared: 2 turns/day (0.27mm/day)
- Please indicate how long the expansion continued for.
- Line 114: Typo. Please remove “(“ and add a “.” At the end of the sentence.
- Line 116: Typo “Scheletal level.”
Figure 6: Please align images of cbct. Diastema measurement in cross-sectional view appears palatalized. Is this intentional?
Figure 7: Please correct for white rectangles in the upper left of each image. Please indicate with arrows where the perforations were made.
Figure 8: Typo in the figure legend. “Splint” should read “Split”
Figure 9: Redundant. Consolidate Figures 8 & 9.
Figure 10 should be referenced sooner when the description of measurements was made in the text.
- Please correct blurry images and crop and align photos into proper figure schematic.
Figure 11: Redundant. Consolidate with Figure 10.
Line 146: Please re-word “In order to determine the type of movement that might have occurred at 1st molar position” to “In order to determine the direction of movement that has occurred at the 1st molar position”
Figure 12: Images should be side by side rather than stacked and cropped to the same size.
Statistical analysis
- Please italicize “p”
- Please provide a range for measurements next to the averages provided
- Line 165-166 “On average, one half of the anterior nasal spine moved more than the contralateral one by 0.89 mm. (p>0.05)”
- Can this conclusion be made given the non-significant p-value?
- Table 2: Please correct inconsistent capitalization of words
- Table 3: Would be better represented by a curve showing distribution. The curve should be labeled to indicate all values indicated in the chart.
- Table 4: Would be better represented by a curve showing distribution. The curve should be labeled to indicate all values indicated in the chart.
- All tables: Please correct for inadequate or erroneous cropping, and please correct for blurry image quality.
Discussion:
- Line 196: Typo. Remove “(“
- Line 200: Unsure why there is “(!)”
- Lines 205-207: Is this a statement true given the comparison has p>0.05? Why or why not?
- Lines 225-228: Reword to read “The limitations of our study include…”
Conclusions
- Conclusion #2: Again- can this be stated?
General comments:
- Interesting concept and use of corticotomies for rapid expansion. Is there any evidence indicating that corticotomies increase the speed of palatal expansion, given that all expansions were performed when the mid palatal suture has not closed?
- Have any other studies been done where a) No implants, but with corticotomies or b) Corticotomies, but without implants, that this data can be compared with?
- How do the speed, amount of spacing, and the nature of the “split” compare with conventional palatal expanders? Is there a benefit for this method, seeing as it is more surgically invasive and possibly costlier?
- When are implants removed? At what timeline? Are there any complications or risks of unintentional osseointegration?
- What implant system was used for this experiment?
- All figures should be “cleaned up” to appear more esthetic and also easier to follow. The sizing of images should be made to have a cohesive appearance. Areas, where patient information was removed, should be filled in to match the background (i.e. White squares in CBCT)
Reviewer 4 Report
1.the authors should define the time frame of the study : what they mean with the "subsequent stage of the study" : the second CBCT was taken at the end of expansion ? at the end of orthodontic treatment? during orthodontic treatment.
I would like to know at what stage of the treatment protocol were the data collected.
2. one of the key point of maxillary expansion in adult patients is stability : no data are provided about stability.
3. In the discussion the authors should mention the option of surgically assisted rapid maxillary expansion with bone borne appliances. The results show that bigger amount of expansion can be achieved with this method. authors should discuss the indications for different technicques based on the entity of discrepancy. the authors should define the indications for MARPE with the proposed method
4.in line 23 of the abstract there is a statement about a positive effect on maxillary advancement: this statement is not proved by any measurements provided in the study. i think this statement should be avoided.
5. image 11 shows very few bone on the medial aspect of the frontal incisor. i would like to know some datas about periodontal effect and complications. may be this go beyond of the scope of the study.
Reviewer 5 Report
Thanks, Authors to choose Applied Sciences and MDPI to publish for MS.
Abstract and Introduction good frame the topic of this research.
Material and Methods well describe the protocol and there is a good iconography.
Please check minor error in results like in the layout of the tables but especially in table 4.
Look these article for a better scientific soundless
PMID: 32900389
PMID: 33378488
Good Job.
Round 2
Reviewer 1 Report
The authors appropriately responded the suggestions from this reviewer.
Reviewer 2 Report
The manuscript was improved greatly.
However, the following issue that I have raised in the previous review was not solved.
"To prove the effect of cortico-puncture, the comparsion between two MARPE groups with cortico-puncture and without corticopuncture should be compared. Otherwise, this study only proves that cortico-puncture do not interfere with the MARPE, and this is meaningless."
Reviewer 3 Report
The authors have addressed the reviewer's comments satisfactorily.
This manuscript is a resubmission of an earlier submission. The following is a list of the peer review reports and author responses from that submission.
Round 1
Reviewer 1 Report
this is an excellent paper from all aspects of Professionalism!
Reviewer 2 Report
In the present study the authors present a case series of patients treated with micro implants assisted palatal expansion (MARPE). The treatment modality is novel and studies on this topic are welcome. In this specific case the methodology and the study design are not fully stated, the study group is not clearly identified. The discussion section needs to be improved. Text editing is needed.
The title itself is not clear clinical studies regarding the advantages make no sense. Why is it plural?
line 29-30 What does it means special elasticity?
Line 31 What are the post-interventional complications associated with RME?
Line 38 seems a case series is it correct?
Figure 1 needs to be flipped vertically
General consideration of the material and methods section: No demographic data were presented, the study group was not defined in terms of age and sex, these data should be available and reported
Line 44 Split mouth means that you are applying a protocol on one side and another one on the opposite side. Doesn't make sense in this case since both sides are affected by the same treatment
line 47 52 the style of the list is not consistent dots and lines are present and in different colour
line 51 how do you assess the asymmetry? the protocol should be stated
Line 54 poor oral hygiene is a too generic term
Line 56 same consideration than in line 54
Figure 2 must be flipped a figure with less cement spread would be appreciated
Line 83 State the clinical objective please
Line 88 if the CBCT was performed after the application of the appliance the selection criteria stage E, sutural fusion as classified by Angelieri et al.[8] has occurred in the maxilla. make no sense since it could not be assessed prior the selection it seems a retrospective study then.
You say in line 72-73 Perforations were made at approximatively 2 mm apart at a depth ranging from 2 mm to 5 mm, depth 72 appreciated to be safe on CBCT examination before surgical treatment but you don't mention a pre treatment CBCT
Figure 7 figure legends are poor and not self explanatory is split not splinting please thing having the text reviewed by a professional english editor
line 155 the Graph is not very useful since repeats exactly the information in the previous table without adding any highlights. milimeters should be millimeters please use the word corrector prior to the submission
Table 2
Significantly different? (P < 0.05) |
Yes |
is unuseful since p was 0.0008 please try to use a normal statistical report protocol
Same consideration for table and graph 2
Table 4 is formatted in a different way than the previous ones you should use a consistent style
line 179 Is it not clear what you mean with typical
In the discussion your findings should be compared with the ones of other authors and this was to done at all
Conclusion should be consistent with the results of this study and of the previous ones that are not mentioned in the discussion part
Reviewer 3 Report
This study was designed as an observational imaging study, regarding 12 structural bone changes that may occur during healing after the placement of micro-implant assisted 13 rapid palatal expander (MARPE) in combination with cortico-puncture therapy (CP).
The use of abbreviations and spacing is not well trimmed, so I feel that the authors wrote it in a hurry.
There are too many figures. Also, as I follow the figures, I think that this study is just a case report.
Figure legends seem to need to be described in more detail.
Tables should be rearranged to make them understandable.
MARPE associated with cortico-puncture therapy may have caused various biological problems, especially in the area of the midpalatal suture. It is regrettable that there is no mention of this.
Reviewer 4 Report
This article is of average interest to the readers. The introduction and the research design can be improved. The methods, results and conclusions are clearly presented and accurate.